# Using Sodium Polyacrylate to Gel-Spin Lignin/Poly(Vinyl Alcohol) Fiber at High Lignin Content

**DOI:** 10.3390/polym14132736

**Published:** 2022-07-04

**Authors:** Manik Chandra Biswas, Ericka Ford

**Affiliations:** 1Department of Textile Engineering, Chemistry and Science, North Carolina State University, 1020 Main Campus Drive, Raleigh, NC 27695, USA; 2Department of Textile Engineering, Chemistry and Science, The Nonwovens Institute, North Carolina State University, 1020 Main Campus Drive, Raleigh, NC 27695, USA; enford@ncsu.edu

**Keywords:** bio-based precursor, lignin/PVA fiber, green/eco-composites, process enhancement, gel aging elimination, continuous process

## Abstract

Lignin is the world’s most naturally abundant aromatic polymer, which makes it a sustainable raw material for engineered polymers and fiber manufacturing. Dry-jet gel-spinning was used to fabricate poly(vinyl alcohol) (PVA) fibers having 30% or more of the lignin biopolymer. To achieve this goal, 0.45 wt.% of aqueous sodium polyacrylate (SPA, at 0.55 wt.% solids) was added to spinning dopes of PVA dissolved in dimethylsulfoxide (DMSO). SPA served to enable the spinning of fibers having high lignin content (i.e., above 30%) while eliminating the aging of as-spun gel fiber prior to elevated temperature drawing. SPA impedes the migration of acetone soluble lignin from the skin of as-spun gel fibers, because SPA is insoluble in acetone, which is also a nonsolvent coagulant for PVA. PVA fibers having 30% lignin exhibited the highest tenacity of 1.3 cN/dtex (centinewton/decitex) and specific modulus 35.7 cN/dtex. The drawn fiber of 70% lignin to PVA, showed tenacity and specific modulus values of 0.94 cN/dtex and 35.3 cN/dtex, respectively. Fourier Transform Infrared (FTIR) spectroscopy showed evidence of hydrogen bonding between lignin and PVA among the drawn fibers. The modification of PVA/lignin dopes with SPA, therefore, allowed for the fabrication of gel-spun biobased fibers without the previously required step of gel aging.

## 1. Introduction 

Recent efforts to meet the emerging demand for high-performance fibers and industrial fibers have focused on the development of high-volume and high-end products from biopolymers. Industrial fibers from natural resources are competitive alternatives of petroleum-sourced synthetic fibers due to high abundancies, biodegradability, non-toxicity, and low cost [1,2,3]. Lignin, the most abundant aromatic biopolymer and waste by-product of pulp and paper mills, showed promising performance as a biorenewable alternative to petroleum-based, expensive raw materials for carbon fiber manufacturing [4,5]. However, lignin has a three-dimensionally branched structure, amorphous nature, and low modulus that limits its applications as a filler material and industrial resin for fiber spinning.

Unlike long chain biosourced polymers, like cellulose, lignin lacks the extensional viscosity necessary for conversion into fibers through solution-spinning. Therefore, lignin is blended with synthetic polymers to achieve fiber through various conventional techniques [4,6]. The polar nature of lignin and its ability to hydrogen bond with polar polymers allow for homogeneous blends [2,7,8]. Kadla et al., reported the production of polyethylene oxide (PEO) and hardwood kraft lignin blended fibers and enhanced physical properties at 5–10 wt% PEO [6]. They found 5 wt% PEO loaded fibers exhibited excellent mechanical properties, i.e., the tensile modulus and strength increased by 10%. According to Awal and Sain, the addition of PEO at 5–20 wt%, plasticized the melt spinning of soda hardwood lignin (SHL). The melt spinning temperature reduced by 10 °C at 20 wt% PEO and observed higher drawing speeds when compared to the neat SHL system- but did not mention spinning draw ratio [2,3]. 

Poly(vinyl alcohol) (PVA), a low-cost biodegradable polymer, is often used as concrete reinforcement due to hydroxyl groups that enhance molecular adhesion between the concrete and fibers. Kubo and Kadla observed strong hydrogen bonding interaction between lignin and PVA during the fabrication of melt spun fibers, exhibiting PVAs compatibility with lignin [9]. However, melt-spun lignin fibers encountered various technical challenges, including phase separation [4,5], brittleness [10], high thermal and chemical stability, etc. [10,11]. A probable way to overcome these challenges is to dissolve the blend in an appropriate solvent and then extrude the blended fibers via solution-spinning. The prime advantage of solution-spun fibers is that solution spinning helps to improve blending when a good solvent can dissolve both polymers. Lu et al. were cable of achieving high strength lignin/PVA fibers using gel-spinning techniques and retaining up to 50% lignin to PVA matrix polymer [12,13]. Tensile strength and Young’s modulus values were 1.1 GPa and 36 GPa, respectively at 5% lignin loadings. In the same study, spinning dopes containing more than 20% lignin resulted in aggregation, the poor alignment of lignin’s aromatic groups along the fiber axis, and stymied the fabrication of strong lignin/PVA fibers using a methanol bath [13]. However, the acetone/methanol coagulation bath overcame those drawbacks [14]. Gel aging plays an important role in fabricating high strength PVA fibers. Tanigami et al. Observed how the aging of PVA gels in low temperature water (up to 120 days at 10 °C) enhanced processing and low-melting-temperature crystals formed [15]. These crystals were unfolded and recrystallized during high temperature thermal drawing at higher draw ratios, which resulted in higher orientation and thereby high tensile strength (2.8 GPa) and modulus (72 GPa) values. For gel-spun PVA fibers, researchers have shown that the tensile properties are improved at a longer gelation time [16]. Lu et al. applied the gel aging method towards the gel-spinning of lignin/PVA fibers and investigated the effect of aging bath solvent and aging time on the processing, structure and properties of the fabricated fibers [12]. The fibers aged in 25/75 water/acetone, dopes at 30% lignin, exhibited higher compatibility, improved draw ratios, and higher alignment of lignin within drawn fibers compared to the 15/85 water/methanol bath at a lower gel aging time of 1 day. At 5% lignin, the lignin/PVA fibers exhibited the highest values of tensile strength (1.3 GPa) and Young’s modulus (50 GPa) from 1 day of gel aging, whereas 1.4 GPa in tensile strength and 54 GPa in Young’s modulus were observed from 14 days of gel aging fibers in water/acetone. However, tensile strength and modulus values decreased to 0.9 GPa and 27 GPa after 30 days of gel aging. This can be explained by the excessive solvation of PVA chains with water- preventing the formation of dense fiber structure as well as lignin leaching into the bath.

The lengthy time period for gel-fiber aging also makes this process unconducive to the continuous fabrication of fiber at scales larger than that of lab-scale. Methods for eliminating gel aging to support the continuous gel-spinning of fiber is absent from peer-reviewed literature to the authors’ knowledge. Aqueous poly(acrylic acid) sodium salt (SPA) was added to gel-spinning dopes of lignin/PVA to yield fibers having higher lignin content and high temperature drawing without aging the as-spun gel fiber. In spite of being a synthetic polymer, SPA’s water-solubility, biodegradability, and non-toxicity allow for its use as a polyelectrolyte and potential additive. Aging is used to develop the gel structure so that it can endure elevated temperature drawing without melting the gel structure. The novelty comes from (1) the use of sodium polyacrylate (SPA) to increase lignin loading into gel fibers. (2) by incorporating SPA, fibers were thermally drawn immediately after the as-spun gel structure was spun. SPA hinders lignin migration and promotes coagulation while maintaining lignin’s concentration in the solvent rich domains. Therefore, the work in this study illustrates a simple and continuous gel-spinning approach that retains lignin in high concentrations and eliminates the need for the laborious gel aging step.

## 2. Experimental Section

### 2.1. Materials

Atactic poly(vinyl alcohol) (a-PVA) (molecular weight 146–186 kg mol^−1^, 99% hydrolysis), dimethyl sulfoxide (DMSO), and aqueous SPA Mw ~8000 g/mol (at 0.55 wt.% solids in H_2_O) were purchased from Sigma Aldrich. Acetone was purchased from BDH Chemicals. All materials were used as received.

Kraft lignin was acquired from Hinton Pulp in Alberta, Canada. This work focused on techniques to improve the lignin loading among solution spun fibers using kraft lignin for which this approach prevents leaching from a saturated gel-fiber. Higher molecular weights would decrease the number of lignin molecules. Nevertheless, this technique is directed towards loading, because loading is influenced by lignin’s saturation of the PVA-rich regions of the as-spun gel fiber followed by the solvent-rich regions. The impurity levels are usually <2% ash content which would not interfere with gelation or fiber drawing.

Kraft lignin is the most commonly available type of lignin; it makes up most of the industrial waste, and its main monomer unit is coniferyl alcohol. Impurities will affect the actual amount of lignin loaded into the fiber, since both lignin and impurities contribute to the total weight. However, our study of lignin diffusion from gel fiber was not influenced by ash content.

### 2.2. Spinning Dope Preparation

Spinning dopes were prepared at 20 g/dL of the lignin/PVA blend dissolved in solvent. Dope containing 30 wt.% lignin and 70% wt.% PVA was termed as Lig3/PVA7. Similarly, Lig5/PVA5 and Lig7/PVA3 named fiber samples at weight ratios of 50/50 and 70/30 (*w/w*) lignin and PVA, respectively. All lignin/PVA polymer dopes contain a constant weight percent of aqueous SPA at 0.45 wt% to lignin/PVA. The Supporting Information describes the manufacturing and processing conditions for fiber without lignin or SPA, which was called neat PVA fiber. Preparation of neat PVA polymer dope and the gel-spinning of PVA fibers are discussed in Appendix A as PVA gel-spinning was done in a methanol coagulation bath at −25 °C and aged overnight.

Spinning dopes of lignin/PVA were prepared from blends of PVA/DMSO and lignin/DMSO. Both polymers were dissolved separately in DMSO at 85 °C, then aqueous SPA was stirred in to ensure homogeneous mixtures (Figure 1a,b). Once the homogeneity of the polymer solution was confirmed through optical micrographs, absent of aggregates, solutions were ready for spinning (50% lignin to PVA is shown in Figure 1b).

### 2.3. Gel-Spinning of Lignin/PVA Fibers

Gel-spinning involves 3 major steps, as illustrated in Figure 1. First, polymer solution that appears dark brown in color (Figure 1a) is loaded into a high-pressure steel syringe and fiber is extruded through a 22-gauge (0.43 mm inner diameter) syringe needle. In Step 1, the air gap was set at 10–12 mm between the syringe tip and coagulation bath (see Figure 1c). Pure acetone was used in the coagulation bath at room temperature. After solvent extraction, as-spun fiber was collected onto a rotating winder. In Step 2, gel aging in a solvent bath was omitted and as-spun fiber was drawn over hot air. Step 3 employs elevated temperatures to point draw fiber from Step 2 through four series of silicone oil at consecutively higher temperatures until the fibers are fully drawn. The draw ratio (DR) for each stage was calculated by Equation (1)
DR = V_2_/V_1_(1)
where, V_1_ and V_2_ represent the velocity of feeding and take-up winders, respectively (Figure 1, Step 2 and 3).

### 2.4. Gel Melting

The thermoreversible nature of PVA gels returns them into liquids of dissolved polymer at elevated temperatures. The effects of lignin content and SPA on the gel melting temperature of thermoreversible gels were measured according to the method of Lu et al. [14]. Dopes were suctioned into the capillary tube (Figure 2b), keeping one end open and the other end capped with Teflon tape (Figure 2). The uncapped capillary end was placed in acetone at a low temperature until the dope gelled. Afterwards, the capillary tube was kept upside down in a test tube, alongside a thermocouple probe. The test tube was propped up in a Thiele tube filled with glycerol (as seen in Figure 2a,b). The flame of the Bunsen burner below the Thiele was used to gradually increase the oil temperature and to induce gel melting. Gel melting point was defined at the temperature where the gel starts to flow (Figure 2c).

### 2.5. Mechanical Testing

The fully drawn fibers were washed with isopropyl alcohol (IPA) to remove residual oil from the fibers’ surface, dried at room temperature for ~24–48 h, and then conditioned overnight at 25 °C and 65% relative humidity prior to testing. Afterwards, the tensile properties of fully drawn fibers were measured using the MTS-Q mechanical testing system according to ASTM D3822. TestXpert data acquisition software was used to process load-displacement data. Mechanical testing was performed with a 25 mm gauge length, crosshead speed of 15 mm/min, and 4.45 N load cell. At least 10 specimen per sample were tested. Fiber toughness was determined from the area beneath the stress-strain curve. Toughness reflects the energy absorbed before mechanical failure. Tensile toughness of each fiber was calculated from the integration of stress–strain curves. It is the energy absorbed until the fiber breaks [14,17]. Fiber mechanical strength under wet conditions was measured for lignin/PVA gel-spun fibers according to ASTM D3822 and after preconditioning according to ASTM D1776.

### 2.6. Analysis of Fiber Microscopy

Lignin/PVA fibers’ cross-sections were imaged on the LEXT OSL4000 3D confocal laser microscope. The lignin/PVA fibers were embedded in a synthetic cork, using trilobal nylon fibers as filler, in preparation for the cross-sectional imaging of lignin/PVA fiber.

The fibers’ morphology at the facture tip of tensile tested samples was performed using scanning electron microscopy (SEM) on the FEI Verios 460L. Samples fractured in tension were carbon taped onto the sample tab, then sputter coated with 60/40 (*w/w*) gold/palladium, and imaged with 2 kV accelerating voltage.

### 2.7. Fiber Structural Analysis by Diffraction and IR Spectroscopy

Wide-angle X-ray diffraction (WAXD) was carried out to measure the crystallinity of fully drawn fibers. The X-ray diffractogram of fully drawn fibers was collected on the Rigaku Smartlab X-ray diffractometer using Cu Kα radiation (λ = 1.541 Å), voltage of 40 kV, and operating current of 44 mA. The highly aligned fiber bundle of 20−30 fibers was scanned at a step size of 0.05° between 5° and 50° 2θ. WAXD patterns were normalized according to Equation (2). The normalized intensity (I) is the ratio of the intensity, i_n_, at 2θ and the sum of intensities over the entire 2θ range.
(2)$$\mathrm{Normalized}\;\mathrm{Intensity}\;(I)\;=\;\frac{i_{n}}{\sum_{n}^{\infty }i}\;\times \;10^{6}$$

Attenuated Total Reflectance (ATR)-FTIR spectra of neat PVA fibers and lignin/PVA were acquired from 4000 to 400 cm^−1^ by NICOLET iS50 spectrophotometer using 128 scans and 4 cm^−1^ spectral resolution. To normalize IR absorbance spectra, the 842 cm^−1^ (C-C stretching) peak was used as the reference band.

### 2.8. Moisture Sensitivity Measurement

The dissolution of lignin/PVA fibers at different concentrations of lignin were imaged by optical microscopy equipped with a Mettler F90 Central Processor and Mettler FP82HT Hot Stage. Sample slides comprised of a longitudinal mount of fiber in the presence of a water droplet (~20 µL). Under the hot stage, samples were heated from room temperature to 85 °C at 10 °C/min. Micrographs were collected at regular intervals to track the resistance or susceptibility of fibers to dissolution.

## 3. Results and Discussion

### 3.1. Effect of SPA on Lignin Retention in Lignin/PVA Gel Structure

In the absence of SPA, the spinning dope of 50% lignin to PVA exhibited the leaching of lignin from gel coagulated in acetone, as shown in (Figure 3a). The brown discoloration of solvent between these photographs in Figure 3 is indicative of lignin migration into acetone from the PVA gel. However, the inclusion of SPA greatly reduced lignin migration into the acetone bath (Figure 3b).

Figure 3 shows negligible amounts of lignin leached into the acetone from lignin/PVA gel fiber (at 50% lignin) containing SPA. But, lignin/PVA without SPA shows lignin diffusion into the acetone bath at low (30% lignin) to high (70% lignin) amounts of lignin (see Figure 3a having 50% lignin to PVA). Darkening of the solvent was indicative of lignin migration into acetone from the gel polymer. Lu et al. proposed gel structures of lignin/PVA fiber at low to high lignin loadings, wherein polymer dope transition into polymer-rich and solvent-rich domains upon gelation [13]. At higher lignin loadings, saturation of polymer-rich domains occurs, and lignin readily migrates into the methanol coagulation bath from solvent-rich domains [13]. The aqueous SPA might reside in solvent-rich domains which allows lignin to stay within as SPA precipitates in acetone. Therefore, the 100% acetone bath restricts lignin’s migration from both the polymer-rich and solvent-rich domains and into the coagulation bath. This results in higher lignin retention within the gel structure (Figure 3b).

### 3.2. Effect of SPA on Lignin/PVA Gel Melting

Gelation occurred as semi-crystalline PVA forms interconnected, polymer-rich domains that are swollen with DMSO. Based on data shown in Figure 4, lignin increases the melting temperature of PVA gels regardless of the presence or absence of SPA. Therefore, lignin must reside in the polymer-rich domains. Lu et al. also observed higher gel melting temperatures among lignin/PVA gels when the 15/85 methanol/acetone coagulation bath induced the preferential residence of lignin among the PVA-rich domains [13]. Lignin can improve the thermal resistance of the PVA gel overall [13,14]; however, any reduction in the degree of semicrystalline PVA gelation can reduce gel melting temperatures.

In this study, gel melting point slightly reduced at 70% lignin content to fiber (Figure 4), which suggests this high concentration of lignin slightly hampered PVA crystallization. Adding SPA to the lignin/PVA gels reduced gel melting temperatures by up to 15 °C (Figure 4). The polyelectrolyte appears to disrupt the crystallization of PVA during gelation. For the remainder of this study, lignin/PVA spinning dopes (Lig3/PVA7, Lig5/PVA5, and Lig7/PVA3) all contain SPA.

### 3.3. Processing Condition of Lignin/PVA Gel-Spun Fibers

The effect of lignin content on the spinning and drawing of lignin/PVA fibers is summarized in Table 1. As-spun fibers were drawn over five stages of drawing. Thermal processing conditions can vary by ±5 °C, and conditions for processing were based on the ability to spin at least 100–200 m of continuous fiber.

Confocal images reveal the cross-sections of additive-modified fibers (Figure 5). The diameter of lignin/PVA fiber are larger compared to neat PVA fibers (Table 1 and Appendix A). This is due to the addition of kraft lignin in PVA, which effects fiber drawing as seen by total draw ratios (Table 1).

#### Mechanical Properties of Lignin/PVA Fibers

The influence of lignin content on the mechanical properties of PVA fibers is shown in Figure 6. Lig3/PVA7 fibers had the highest tenacity of 1.3 cN/dtex (Table 2). Interestingly, drawn fiber at 30 and 70% lignin had similar values of specific modulus (35.7 and 35.3 cN/dtex, respectively). However, the Lig7/PVA3 fibers had a lower tenacity of 0.94 cN/dtex. Still impressive, these are the highest percentages of lignin loading ever reported among gel-spun lignin/PVA fibers.

Water is known to reduce the mechanical properties of PVA fibers, and so, the effects of lignin on the wet strength of fibers was tested. In all cases, the wet tenacity and wet specific modulus of lignin/PVA fibers were significantly weaker than for the dry condition (Table 2). Lig3/PVA7 fibers exhibited the highest retention of tenacity (52%) after wetting. In contrast, Lig7/PVA3 fibers retained ~40% of their tenacity upon wetting (Table 2).

Based on SEM images of fracture tips (Figure 7), lignin causes the fibers to exhibit brittle failure modes. Nevertheless, the fracture of Lig3/PVA7 does show some necking prior to failure. Further Lig7/PVA3 fibers, show fibrils of PVA that seems to reside in a matrix of less organized polymer. The less organized polymer is deemed to be lignin.

### 3.4. Effect of Lignin on Fiber Microstructure

WAXD was used to study structural changes among fully drawn fibers. The most prominent diffraction peak in Figure 8 is due to the overlapping of (10$\bar{1}$) and (101) peaks at 2θ = 19.6°. Lignin can disrupt the crystallization of PVA in drawn fibers as noted by the decrease of (10$\bar{1}$) peak intensity. Amorphous scattering is more prominent as lignin content increases from 30 to 70% of the PVA fiber. According to IR, H-bonding between lignin and PVA results in the disruption of PVA-PVA bonding leading to a reduction in PVA’s percent crystallinity.

Intermolecular bonding between lignin and PVA induces molecular adhesion. The IR absorbance spectrum from 3000 to 3700 cm^−1^ (Figure 9) provides insight into hydrogen bonding for lignin and PVA. Table 3 describes the functional groups characteristic of PVA, lignin, and DMSO as detected by IR spectroscopy. Among lignin/PVA fibers, the absorbance peak of -OH shifted from 3304 cm^−1^ (see SI) for neat PVA to 3284 cm^−1^ for Lig3/PVA7 fiber (Figure 9). This implies hydrogen bonding between lignin and PVA [6,9,18]. At the highest lignin content (70%), the 3284 cm^−1^ peak shifted to 3318 cm^−1^ which suggesting less intermolecular interaction than Lig3/PVA7 fibers. Moreover, the -OH stretching vibration for pure PVA (at 3304 cm^−1^) gradually broadened and weakened, while the C-O stretching vibration at 1087 cm^−1^ increased. Strong peaks at 1261 cm^−1^ are indicative of C-O stretching in the guaiacylic rings, which are derived from the coniferyl lignin monomer found in both hardwood and softwood sources. Also, characteristic of lignin, strong peaks between 798–810 cm^−1^ are indicative of aromatic C-H stretching [1]. The characteristic peak for crystalline conformations of neat PVA (at 1141 cm^−1^) decreased as the lignin loadings keep increasing from 30–70% [18,19]. Amorphous PVA is associated with the C-O vibrational mode at 1087 cm^−1^ and PVA crystallinity affected 1141 cm^−1^ absorbance [18,20].

### 3.5. Moisture Sensitivity

The moisture resistance of lignin/PVA fibers at room and elevated temperatures was tested using optical microscopy. All fibers maintained their geometry after immersion in room temperature water for 5–10 min although swelling was evident among the fibers. At 85 °C in water, the 30% and 50% lignin fibers appeared to be swollen gels (Figure 10). But with 70% lignin, the lignin/PVA fiber disintegrated and dissolved in water (Figure 10). The reduced crystallinity of fibers is believed to permit moisture diffusion, and the fiber dissolution in water. It is also possible that the presence of the SPA polyelectrolyte causes moisture absorption and swelling.

## 4. Conclusions

In this work, lignin/PVA fibers are fabricated with the highest loading of lignin reported to date, creating a low-cost biobased fiber with minimal dependency on synthetic vinyl polymer. An uninterrupted gel-spinning approach is used to spin lignin/PVA fibers having dry tenacity and dry specific modulus values that are competitive to wholly synthetic industrial fibers. Fibers with 30% lignin to PVA exhibited the highest mechanical strength- tenacity 1.3 cN/dtex (5.33 MPa) and tensile modulus 35.7 cN/dtex (146 MPa)- among all the modified fibers. Drawn fibers fabricated from solutions containing 70% lignin, showed tenacity and modulus values of 0.94 cN/dtex (4 MPa) and 35.3 cN/dtex (135MPa), respectively. Although the mechanical properties underperform in lieu of fibers containing 30% lignin; nevertheless the reported properties are notable in consideration of the high loadings of lignin retain within the PVA matrix [8,9,12,13,14]. Aqueous sodium polyacrylate was added to dopes of lignin/PVA to yield high strength fibers without the use of gel aging. This technology eliminates the aging step and makes lignin/PVA gel-spinning amendable to continuous processing. This study to eliminate gel aging when spinning lignin/PVA fibers highlights the interplay between lignin content and processing steps on the ability to increase bio-content among man-made fibers, lower price, and to achieve acceptable performance properties among engineered cementitious composites for the construction industry.

## Figures and Tables

**Figure 1 polymers-14-02736-f001:**
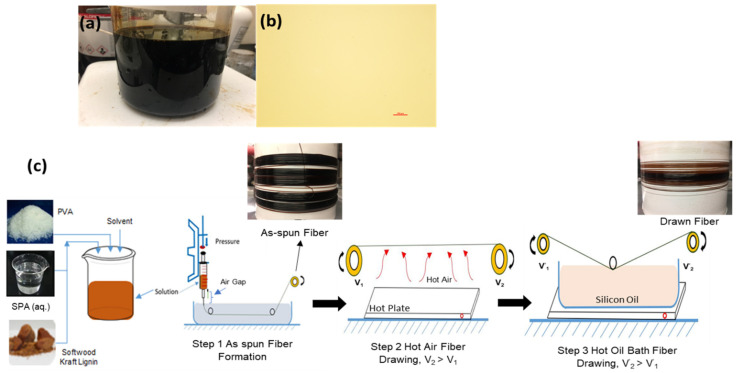
The homogeneity of lignin/PVA dopes is visible within (**a**) digital photograph and (**b**) optical micrograph of Lig5/PVA5 dope; (**c**) schematic for the dry-jet gel-spinning of Lig5/PVA5 fibers shows three processing steps: as-spun fiber formation (Step 1), hot air fiber drawing (Step 2), and hot oil multi-stage fiber drawing (Step 3).

**Figure 2 polymers-14-02736-f002:**
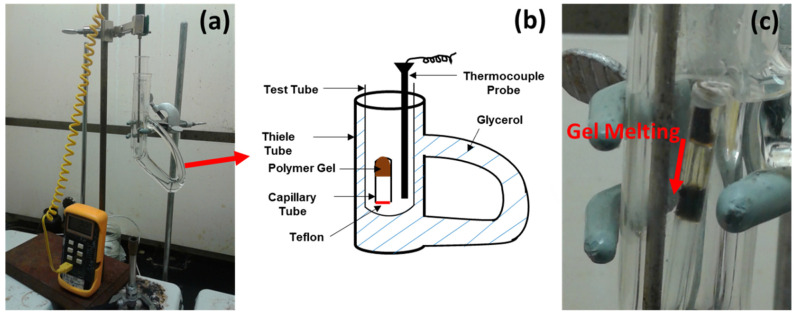
The testing set-up for measuring gel-melting is photographed in (**a**) along with an illustration of the Thiele tube (**b**). The test tube housed the glass capillary tube of gel and the thermocouple to allow for the gradual heating of this inner chamber, as the Thiele tube of glycerol was heated by the Bunsen Burner. Melting of Lignin/PVA gel is demonstrated in (**c**).

**Figure 3 polymers-14-02736-f003:**
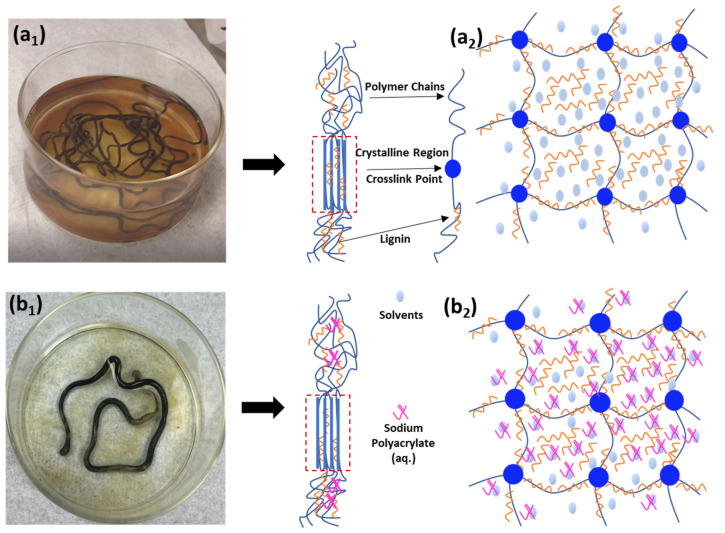
Illustration of fiber microstructures of Lignin/PVA 1:1 (*w/w*) fibers (**a**) without SPA additive and (**b**) with SPA additive at 50% lignin content in (**1**) digital photographs of coagulation test and (**2**) fiber microstructures.

**Figure 4 polymers-14-02736-f004:**
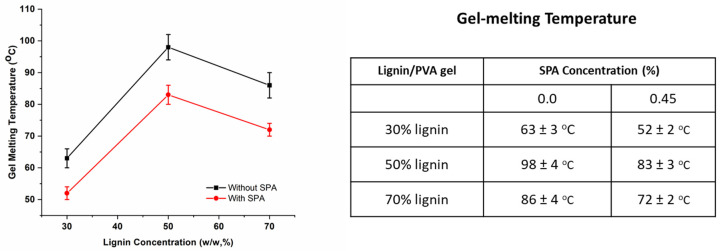
Gel-melting point of lignin/PVA with (red circles) and without (black squares) aqueous SPA.

**Figure 5 polymers-14-02736-f005:**
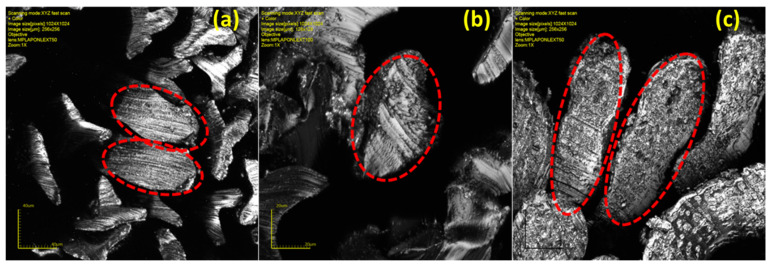
Confocal micrographs of (**a**) Lig3/PVA7, (**b**) Lig5/PVA5, and (**c**) Lig7/PVA3 fibers.

**Figure 6 polymers-14-02736-f006:**
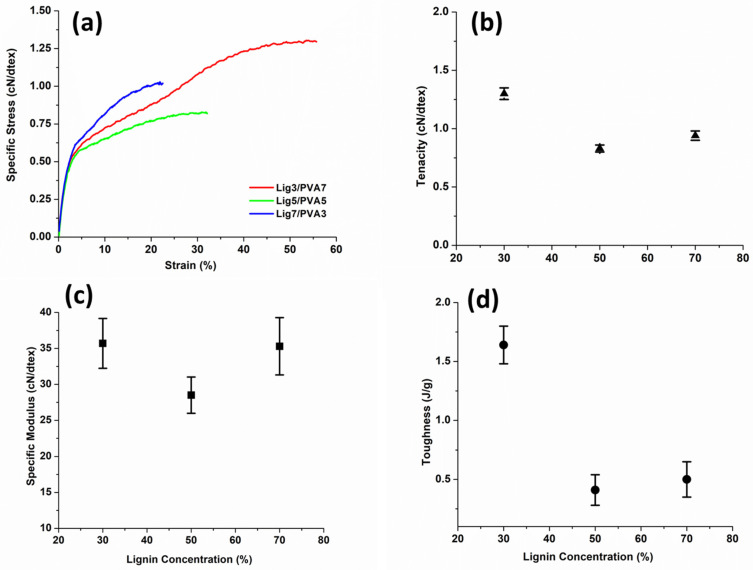
Tensile profile: (**a**) specific stress versus strain, (**b**) tenacity, (**c**) specific modulus, and (**d**) toughness of Lignin/PVA fibers at the dry condition.

**Figure 7 polymers-14-02736-f007:**
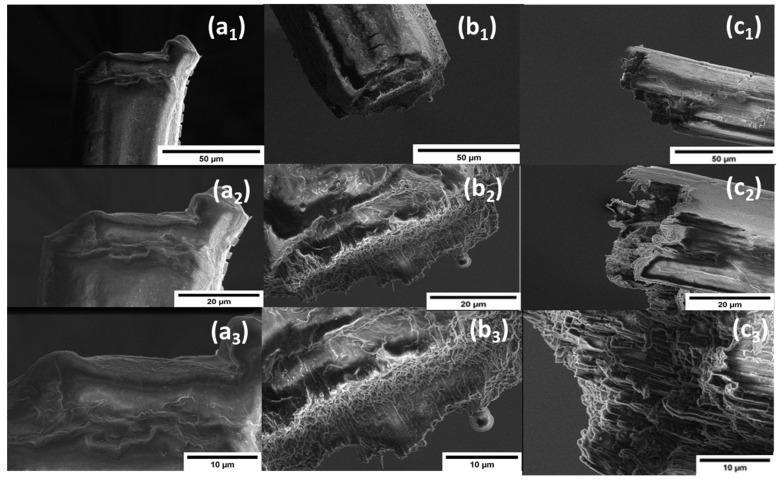
Fracture tips of lignin/PVA fibers: (**a**) Lig3/PVA7, (**b**) Lig5 PVA5, and (**c**) Lig7/PVA3 imaged with SEM at low (1,2) to high (3) resolution.

**Figure 8 polymers-14-02736-f008:**
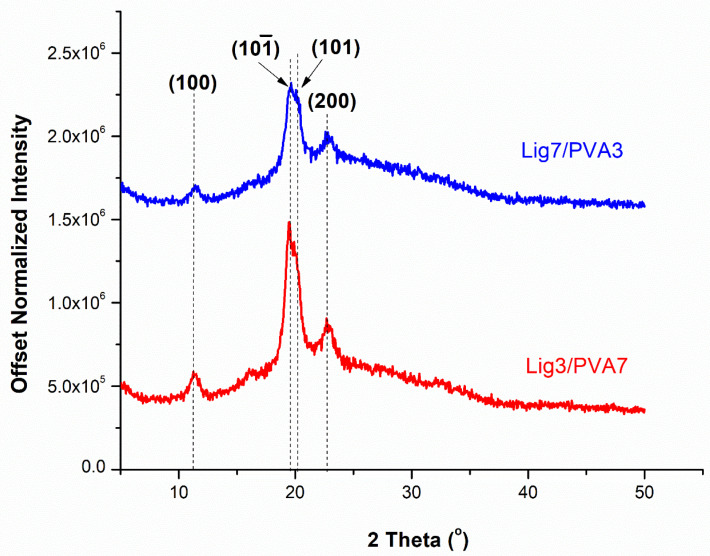
WAXD diffractograms of Lignin/PVA gel fibers.

**Figure 9 polymers-14-02736-f009:**
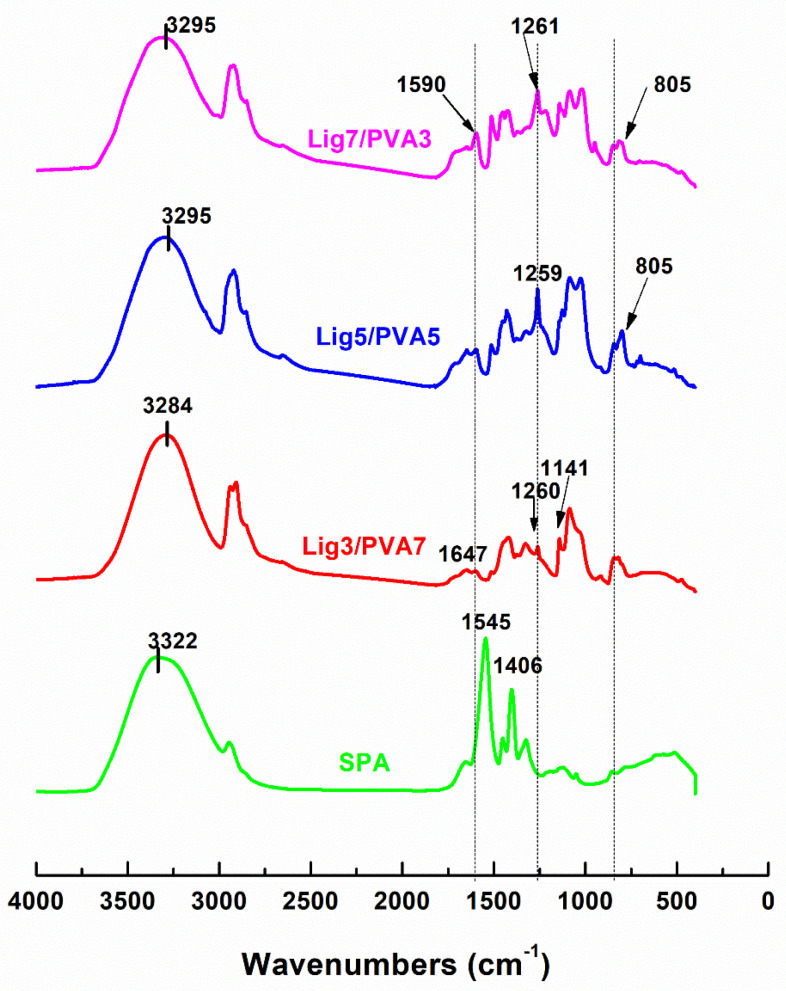
IR absorbance spectra of Lignin/PVA fibers.

**Figure 10 polymers-14-02736-f010:**
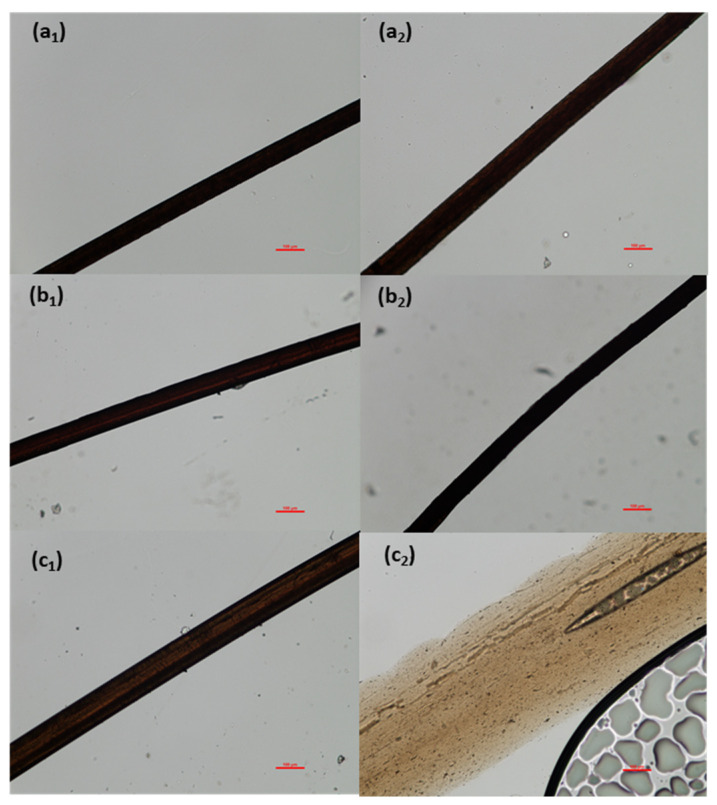
Optical Micrographs of (**a**) Lig3/PVA7, (**b**) Lig5/PVA5, and (**c**) Lig7/PVA3 fibers after immersion in water at (1) 25 °C and (2) 85 °C (100 µm scale bar).

**Table 1 polymers-14-02736-t001:** Drawing Parameters of Gel-Spun Lignin/PVA Fibers at 0. 45% SPA.

Parameters		Samples		
Additive Conc. (%), [0.45% (*w/w*)]		Lig3/PVA7	Lig5/PVA5	Lig7/PVA3
As-Spun Draw Ratio (DR) (with 10 mm Air Gap)		2.5	2	3
Hot Air Drawing	DR	2.7	2.5	2.5
	Temp. ~60 (°C)			
Stage 1 Drawing	Oil Temp. (°C)	90	95	100
	DR	2.0	2.1	1.5
Stage 2 Drawing	Oil Temp. (°C)	135	130	140
	DR	1.7	2	2
Stage 3 Drawing	Oil Temp. (°C)	170	165	180
	DR	1.4	2	1.5
Stage 4 Drawing	Oil Temp. (°C)	225	220	220
	DR	1.6	1.9	1.5
* Total DR		52	48	51
Effective Diameter (µm)		71	76	74
Linear Density (dtex)		28	32	30

* Total DR: Cumulative draw ratio from as-spun DR, Hot-air drawing, and thermally drawing fiber (Stages 1–4).

**Table 2 polymers-14-02736-t002:** Mechanical Properties of Lignin/PVA Fibers.

Sample	Dry Condition				Wet Condition	
	Specific Modulus (cN/Dtex)	Tenacity (cN/Dtex)	Strain at Break, (%)	Toughness J/g	Specific Modulus (cN/Dtex)	Tenacity (cN/Dtex)
Lig3/PVA7	35.70 ± 3.47	1.30 ± 0.05	55.73 ± 8.58	0.52 ± 0.05	17.42 ± 1.96	0.50 ± 0.03
Lig5/PVA5	28.50 ± 2.52	0.83 ± 0.03	32.10 ± 2.01	0.15 ± 0.06	18.00 ± 1.41	0.70 ± 0.05
Lig7/PVA3	35.30 ± 3.98	0.94 ± 0.04	22.50 ± 4.96	0.17 ± 0.03	11.29 ± 1.8	0.75 ± 0.04

**Table 3 polymers-14-02736-t003:** Identification of FTIR peaks in Lignin/PVA/SPA blend Fibers.

IR Frequency (cm^−1^)	Peak Identification & Bond Assignment	Vibration Mode	References
3000–3700	-OH	-OH stretching from water, PVA, and hydrogen bonding between -OH groups.	[14,19,21]
2923	-C-H (CH_2_)	Stretching asymmetric	
1655–1665	(H-O-H) Water bending	Bending vibration	[22,23]
1456	-CH_2_-	-CH_2_ stretching and possibly -OH bending	[18]
1256–1265	-CO stretching	indicative of -CO stretching in the guaiacylic rings present in lignin	[7]
1141–1144, 1155	PVA chain	Indicative of the conformation of PVA chains in the crystalline region.	[19,20,24,25]
1087–1120	PVA chain	Indicative of the conformation of PVA chains in the amorphous region was affected by solvation.	[19,20,24,25]
1014–1050	-S=O Stretching	-S=O stretching originated from residual DMSO within drawn fibers.	[19,20,24,25]
842	-CH_2_-	Skeletal vibrations	[26]
798–810	-CH stretching	indicative of aromatic C-H stretching	[7]

## Data Availability

Not applicable.

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
