# Peer review of "Using Sodium Polyacrylate to Gel-Spin Lignin/Poly(Vinyl Alcohol) Fiber at High Lignin Content"

_polymers, 2022, doi:10.3390/polym14132736_

Round 1
Reviewer 1 Report
Dear Authors, thank you for the manuscript submission.
The current paper describes the properties of lignin/PVA fibers spinning with Sodium Polyacrylate addition. This is an interesting manuscript that needs some improvement before being accepted. Some improvements are necessary as described below.
- Introduction section: The essence of the research is to examine the effect of SPA and the aging mixture on the acceleration of the aging process of the obtained fibers. So it would be useful to add information about the other Lignin / PVA mixtures used for aging the fibers. The literature review shows that it can therefore be concluded that replacing water with methanol in the acetone / water mixture will accelerate the aging process of the fibers, could the authors suggest why it is possible?
- Materials and Methods section: Please explain what was the reason for using DMSO as solvent for Kraft Lignin and PVA?
- Please enter the unit g / mol for SPA mass weight. Especially, molecular weight is a fundamental parameter affecting the phys-ical properties.
2.2 Section:
- What was the proportion of solvent and PVA material, and lignin and DMSO? Solubility depends on the molecular weight of both lignin and PVA. Were these solutions saturated at 85 degrees Celsius?
- Due to the difficult analysis of the information contained in the figure, I suggest and please consider dividing the images in Figure 1 into two parts. Show the information contained in 1a and 1b as a separate drawing.
- Please indicate the diameter of the needle tip and the diameter of the PVA / lignin fibers produced by the spinning process.
2.3 Section
- “ …First, polymer solution that appears dark brown in color (Fig. 1a) …” . Repetition of the information discussed a few lines above.
- “ … Step 3 employs elevated temperature to point …” . In the part describing the methodology, I miss information about the process temperatures in individual stages. Please state what were the temperature values and why were the tests carried out at these temperatures?
- “Gel melting point” - What was the accuracy of the gel melting point measurement?
The chapter entitled: "Gel-Spinning of Lignin / PVA Fibers" and the chapter "Gel melting" have the same number (2.3), please correct the chapter numbering or combine them.
- Was there any statistical analysis of the obtained test results? On what basis was the scatter of the results visible as whiskers in the charts determined?
- Fig 9. Why did the authors not introduce IR absorbance spectra of PVA and Kraft lignin, when SPA absorbance spectra appears on the figure 9?
- 3.4 Section. Kraft lignin and PVA were used to produce the fiber - please specify in the methodology what was the solubility of the lignin and PVA in water. Then conduct a discussion and comment on the water sensitivity of the obtained PVA / lignin fibers.
- The effect of SPA seems to reduce the lignin / PVA bond strength and water absorption capacity, this is just my suggestion.
So my decision is: Major revision, which you should make according to the Reviewer's indications.
Best regards
Reviewer
Author Response
- Introduction section: The essence of the research is to examine the effect of SPA and the aging mixture on the acceleration of the aging process of the obtained fibers. So it would be useful to add information about the other Lignin / PVA mixtures used for aging the fibers. The literature review shows that it can therefore be concluded that replacing water with methanol in the acetone/water mixture will accelerate the aging process of the fibers, could the authors suggest why it is possible?
Ans. The literature review covered the relevant Lignin/PVA research.
The introduction section highlighted the effect of aging solvent and time on processing and structure and properties of the fabricated fibers, which are highlighted as blue.
- Materials and Methods section: Please explain what was the reason for using DMSO as a solvent for Kraft Lignin and PVA?
Ans. Dimethyl sulfoxide (DMSO) helps to dissolve both the materials in order to get a homogeneous mixture (Fig. 1b).
- Please enter the unit g / mol for SPA mass weight. Especially, molecular weight is a fundamental parameter affecting the physical properties.
Ans. Corrected as per suggestion.
2.2 Section:
- What was the proportion of solvent and PVA material, and lignin and DMSO? Solubility depends on the molecular weight of both lignin and PVA. Were these solutions saturated at 85 degrees Celsius?
Ans. Spinning dopes were prepared at 20 g/dL of Lignin/PVA blend dissolved in the solvent. Polymer solution containing 30 wt.% lignin, and 70% wt.% PVA was termed as Lig3/PVA7. Similarly, Lig5/PVA5 and Lig7/PVA3 named fiber samples at weight ratios of 50/50 and 70/30 (w/w) lignin and PVA, respectively.
Both polymers were dissolved separately in DMSO at 85 oC, then SPA was stirred to ensure homogeneous mixtures (Fig. 1a, b). Once homogeneity of the polymer solution was confirmed through optical micrographs absent of aggregates, solutions were ready for spinning. of 50% lignin to PVA is shown in Fig. 1b.
- Due to the difficult analysis of the information contained in the figure, I suggest and please consider dividing the images in Figure 1 into two parts. Show the information contained in 1a and 1b as a separate drawing.
Ans. The author greatly appreciates the reviewer’s comment. Here, Fig. 1 a and 1b is to show a homogeneous mixture.
- Please indicate the diameter of the needle tip and the diameter of the PVA / lignin fibers produced by the spinning process.
Ans. The diameter of the needle tip was 0.43 mm inner diameter which is included in the revised version of the manuscript.
The diameter of Lignin/PVA fibers is mentioned in Table 1.
2.3 Section
- “ …First, polymer solution that appears dark brown in color (Fig. 1a) …” . Repetition of the information discussed a few lines above.
- “ … Step 3 employs elevated temperature to point …” . In the part describing the methodology, I miss information about the process temperatures in individual stages. Please state what were the temperature values and why were the tests carried out at these temperatures?
Ans. Step 1 was done at room temperature.
Step 2 was carried out Temp. ~60 °C and Step 3 was carried out in several stages, with temperature ranging from 90-225 °C.
These processing conditions are mentioned in Table 1.
- “Gel melting point” - What was the accuracy of the gel melting point measurement?
Ans. The gel melting point was measured triplet for each sample and highlighted in Fig. 4.
The chapter entitled: "Gel-Spinning of Lignin / PVA Fibers" and the chapter "Gel melting" have the same number (2.3), please correct the chapter numbering or combine them.
Ans. Corrected as per suggestion.
- Was there any statistical analysis of the obtained test results? On what basis was the scatter of the results visible as whiskers in the charts determined?
Ans. The author did not understand which test results the reviewer referred to here.
The author will appreciate it if the reviewer could specify the test results that need to address.
- Fig 9. Why did the authors not introduce IR absorbance spectra of PVA and Kraft lignin, when SPA absorbance spectra appear in figure 9?
Ans. Here, the author just tries to show where there is any interaction evidence of SPA with Lignin or PVA. As lignin and PVA spectrums are well known and here we highlighted their interaction and H-bonding formation.
- 3.4 Section. Kraft lignin and PVA were used to produce the fiber - please specify in the methodology what was the solubility of the lignin and PVA in water. Then conduct a discussion and comment on the water sensitivity of the obtained PVA / lignin fibers.
The effect of SPA seems to reduce the lignin / PVA bond strength and water absorption capacity, this is just my suggestion.
Ans. In section 3.4, the author discussed the leaching behavior of Lignin into acetone.
Reviewer 2 Report
Dear author,
This paper deals with the Using Sodium Polyacrylate to Gel-Spin Lignin/Poly (Vinyl Alcohol) Fiber at High Lignin Content. It is an interesting paper, but several issues must be attended to before being considered for publication.
Abstract section.
Must be more informative in a general way. What is cN/dtext?...define it or use IS units.
Introduction section.
It is too large, consider to shortener. Also, clarify the phrase… [ref. chu hong].
Methodology section.
The molecular weight of the kraft lignin is unknown. This is an unnecessary sentence.
In ATR method… Inter- and intra-molecular hydrogen bonding among lignin and PVA chains were analyzed from IR absorbance in the 3000–3750 cm−1 range. This sentences belongs to discussion section…here the appropriate sentences is specra were recorded from 4000 to 400 cm-1.
Result and Discussion section.
It is important to determine the molecular weight of lignin by GPC.
It is necessary to add a thermal characterization technique (TGA or DSC) to characterize the materials.
It is necessary to make an appropriate discussion of all your results and to compare them with previous works like in the introduction section.
Author Response
Abstract section.
Must be more informative in a general way. What is cN/dtext?...define it or use IS units.
Ans. centinewton/decitex is expressed as cN/dtex and calculated from the breaking force and linear density.
Introduction section.
It is too large, consider to shortener. Also, clarify the phrase… [ref. chu hong].
Ans. The authors believe this is a very important topic to investigate. So, it was necessary to discuss the research question and appropriate background well enough to educate the broader audience. However, the authors tried to concise the introduction as per recommendation.
The phrase is corrected as per recommendation.
Methodology section.
The molecular weight of the kraft lignin is unknown. This is an unnecessary sentence.
Ans. Corrected.
In the ATR method… Inter- and intra-molecular hydrogen bonding among lignin and PVA chains were analyzed from IR absorbance in the 3000–3750 cm−1 range. This sentence belongs to discussion section…here the appropriate sentences is specra were recorded from 4000 to 400 cm-1.
Ans. Corrected as per suggestion.
Result and Discussion section.
It is important to determine the molecular weight of lignin by GPC.
Ans. Kraft lignin was acquired from Hinton Pulp in Alberta, Canada. This work focused on techniques to improve the lignin loading among solution spun fibers using kraft lignin for which this approach of preventing leaching from saturated gel-fiber works. Higher molecular weights would decrease the number of lignin molecules. Nevertheless, this technique is directed towards loading, because loading is influenced by lignin saturation of the PVA-rich regions of the as-spun gel fiber followed by the solvent-rich regions. The impurity levels are usually <2% ash content which would not interfere with anything, also 2% is also low.
Kraft lignin is the most common type, it makes up most of the industrial waste and its main monomer unit is coniferyl alcohol. Impurities will affect the actual amount of lignin loaded into the fiber since both lignin and impurities contribute to the total weight. However, our study of lignin diffusion from gel fiber (see Fig. 3) would not be influenced by ash content.
It is necessary to add a thermal characterization technique (TGA or DSC) to characterize the materials.
Ans. This work focused on techniques to improve the lignin loading among solution spun fibers using kraft lignin for which this approach of preventing leaching from saturated gel-fiber works. Also, the goal was to fabricate textile-grade fiber formation without a long-term gel aging process at higher lignin loading. That’s why the author mainly characterized the tensile properties of the final drawn fiber and highlighted the reasons behind the property improvement.
It is necessary to make an appropriate discussion of all your results and to compare them with previous works like in the introduction section.
Ans. The authors really appreciate the reviewer’s insight about results comparison with literature. The authors believe the results and discussion sections are well backed-up with previous work on similar research and all the references are highlighted in blue color.
Round 2
Reviewer 1 Report
Dear Authors, thank you for revising the manuscript in line with my comments. I consider manuscrypt ready for publication.
Best regards
Reviewer
Author Response
I thank you for your time and support in providing valuable suggestions for the suitability of the manuscript for publication.
Reviewer 2 Report
Dear authors,
This work still needs a proper comparison with previous results and discussion on lignin- PVA fiber's mechanical properties. For instance, The sentence ... Still impressive as it is the highest percentages of lignin loading ever reported in gel spun lignin/PVA fibers....lack a comparison references value...based on what you state this?. I think improving discussion on this topic will realm the value of the present work.
Author Response
The author truly appreciates the reviewer's comment here.
The revised manuscript includes proper references to back up the statement we made here.
I thank you for your time and support in providing valuable suggestions for the suitability of the manuscript for publication.